# The Impacts of Enlarged Subarachnoid Space on Brain Growth and Cortex Maturation in Very Preterm Infants

**DOI:** 10.3390/diagnostics15172206

**Published:** 2025-08-30

**Authors:** Liangbing Wang, Yubo Zhuo, Fang Lin, Xueqing Wan, Guohui Yang, Jianlong He

**Affiliations:** 1Department of Neonatology, The University of Hong Kong, Shenzhen Hospital, Shenzhen 518053, China; linf@hku-szh.org (F.L.); wanxq@hku-szh.org (X.W.); 2Department of Pharmacy, The University of Hong Kong, Shenzhen Hospital, Shenzhen 518053, China; zhuoyb@hku-szh.org; 3Department of Radiology, The University of Hong Kong, Shenzhen Hospital, Shenzhen 518053, China; yanggh@hku-szh.org (G.Y.); hejl@hku-szh.org (J.H.)

**Keywords:** enlarged subarachnoid space, preterm infants, brain segmentation

## Abstract

**Objectives:** The aim of this study is to investigate the changes in quantitative indices of brain volume and cortex development in preterm infants with enlarged subarachnoid space (ESS). **Methods:** A single-center retrospective cohort study was performed in Hong Kong University–Shenzhen Hospital from November 2014 to November 2023, involving 200 preterm infants whose brain MRI images were available. Parameters including the volume of cerebrospinal fluid (CSF), brain tissues, total intracranial cavity (ICC), and key indices of cortex maturation (surface area, cortical thickness, cortical volume, mean curvature) were compared between the groups with ESS and without ESS. The retrospective nature of this study may introduce selection bias in the process of enrolling preterm infants with ESS. **Results:** The groups with severe and mild ESS had a significantly greater ICC volume than the group without ESS (severe: 384.66 ± 30.33 [*p* < 0.001]; mild: 374.25 ± 26.45 [*p* < 0.001] vs. no ESS: 356.78 ± 26.03), and the difference was mostly due to the gap in extra-CSF volume among the three groups (severe: 74.20 ± 5.1 and mild: 55.36 ± 3.8 vs. no ESS: 40.54 ± 4.3, *p* ≤ 0.001). Only the volume of parenchyma of the severe-ESS group was significantly different (severe: 302.35 ± 26.43 vs. no ESS: 312.27 ± 20.75, *p* = 0.003). Regarding indices of cortex maturation, only the mean curvature showed a significant difference between the three groups, and most of the significant clusters were located around the parietal and temporal lobes. **Conclusions:** ESS may be associated with impaired early brain maturation in preterm infants after birth. A further neurodevelopmental follow-up study is needed.

## 1. Introduction

An enlarged subarachnoid space (ESS) is very common in preterm infants, especially in the low-gestational-age (GA) population, where the incidence is reported to be higher than 40% [1]. ESS in preterm infants is usually considered a physical change because it is not often clinically presented and the enlarged space could resolve with time [1,2]. Recent studies have found that the enlargement of subarachnoid space in preterm infants may not be as benign as initially thought. The results showed that those with ESS had a lower mental development index (MDI) and psychomotor development index (PDI) at 2 years of age than those without ESS among preterm infants aged less than 32 weeks [3,4]. In circumstances where this widening of the extra-cerebral space persists, the probability of self-limiting or persistent developmental delay in childhood was also high [5,6,7]. These new findings highlight the importance of understanding whether ESS is associated with adverse neurological presentations in preterm infants, and studies investigating this should include key aspects such as the precise incidence of ESS at different GA levels, perinatal risk factors, impact on brain maturation, and accurate information on the dynamic change in ESS. However, few studies have focused on those issues.

Regarding the pathophysiology of ESS, it refers to abnormal accumulation of cerebrospinal fluid (CSF) in the extra-brain space, and it is diagnosed when the craniocortical width (CCW) is higher than 4 mm in the coronal view of Moro on brain US or MRI in preterm infants. The increased volume of CSF in extra-cerebral spaces is not always caused by the decreased volume of brain parenchyma [8]; it may also be a continuation of the larger extra-cerebral spaces in the fetus which can rapidly shrink to normal levels within a few days after birth [9,10]. This means that the widening of extracranial spaces may not necessarily be an outcome but could also be a cause. The overaccumulation of CSF could limit the brain growth during a critical window of brain development; however, quantitative analysis focusing on the ESS is still relatively scarce. Moreover, the increased pressure caused by overaccumulation of CSF in the outer space could directly compress the adjacent cortex and consequently impair the normal trajectory of cortex development.

In this study, we aim to explore the associations of ESS with brain growth and cortex maturation using advanced post-processing technology based on neonatal brain MRI. We focus on the quantitative assessment of cerebrospinal fluid, obtaining precise volumes of extra-cerebral spaces and cerebrospinal fluid within the ventricular system through dual-platform segmentation, which addresses the inaccuracies of single-platform cerebrospinal fluid boundary segmentation. We have also utilized the latest neonatal cortical segmentation technology to precisely quantify the development process of the cerebral cortex in preterm infants, obtaining key data on cortical development indicators.

## 2. Methods

### 2.1. Patient Selection and Data Collection

Our study was based on existing MRI data collected in a retrospective cohort study of preterm infants born at <32 weeks gestational age (GA). The preterm infants were admitted to the NICU of the University of Hong Kong-Shenzhen Hospital (HKU-SZH) in Shenzhen, China, between November 2014 and November 2023. This study was approved by HKU-SZH Research Ethics Committee, and written parental consent was obtained before MRI scanning. Because we only used clinically acquired data, informed consent from the parents for study participation was waived by the HKU-SZH Research Ethics Committee.

We enrolled the infants who survived before discharge and had complete data from a good-quality brain MRI. Infants who had major congenital anomalies or chromosomal anomalies were excluded. Infants with grade 3–4 intraventricular hemorrhage (IVH) and cystic periventricular leukomalacia (PVL) were excluded to avoid the possible confounding effects of such lesions during segmentation. The infants who had meningitis or stroke were excluded because these conditions may cause hydrocephalus or brain atrophy, respectively, which may also increase the bias on volumes of cerebrospinal fluid (CSF). Out of 415 preterm infants born weighing <1500 g or at <32 weeks’ gestational age who survived in our NICU during the study period, only 225 had complete MRI data. Infants (*n* = 18) with the following were excluded: severe IVH (*n* = 12), cystic PVL (*n* = 5), fetal ventriculomegaly (*n* = 1); 7 cases were abandoned due to unqualified imaging. The remaining 200 infants were divided into three groups according to the value of CCW (Figure 1). We focused on collecting clinical data related to brain volume and cortex maturation, which included intrauterine growth retardation (IUGR)/postnatal steroid usage/severe bronchopulmonary dysplasia (BPD) and brain injuries which were diagnosed on brain US or MRI, such as grade 1/2 IVH and punctate white matter lesions (PWMLs). Clinical data were collected from electronic medical records and nursing records. Basic clinical characteristics such as gestational age (GA), birth weight, sex, IUGR, and severe BPD were based on a previous definition. Head circumference on the day of MRI scanning was recorded.

### 2.2. Study Design and Rationale

The aim of this study is to investigate the association between ESS and brain development, and a retrospective study design can achieve this purpose. To answer our research question, we applied three strict exclusion criteria. First, we excluded infants with central nervous system diseases which can obviously affect the metabolism of CSF, such as meningitis and fetal ventriculomegaly. Second, we also excluded infants with severe brain injury, as this can introduce a large bias in brain tissue segmentation. Thirdly, we set the range of gestational age at MRI scan (37–40 weeks) to study the interaction time between ESS and brain growth.

### 2.3. MRI Data Acquisition

In our NICU, we suggested a brain MRI for preterm infants with birth weight < 1500 g or GA < 32 weeks; the MRI scanning time is usually arranged at term-equivalent age (TEA). All examinations were supervised by a pediatrician experienced in MRI procedures. Infants were given a dose of chloral hydrate (50 mg/kg) orally before the examination. Pulse oximetry and respiratory rate were measured throughout the MRI examination. Ear protection was used for each infant.

All MR imaging was performed with T1-weighted imaging in axial and sagittal planes and T2-weighted imaging in axial and coronal planes on a Simens Avanto 1.5T MRI system (Siemens Healthcare, Erlangen, Germany). Brain MRI data were acquired using a 20-channel head coil. The MRI protocol consisted of sagittal T1-weighted turbo spin-echo images (echo time [TE], 69 ms; repetition time [TR], 6000 ms; flip angle, 150 degrees; voxel size, 0.7 mm × 0.7 mm × 4 mm; field of view [FOV], 200 mm × 175 mm), coronary T2-weighted images (TE, 79 ms; TR, 5000 ms; TI, 1800 ms; flip angle, 150 degrees; voxel size, 0.7 mm × 0.7 mm × 4 mm; FOV, 200 mm × 175 mm), coronary T1-mpr (TE, 2.91 ms; TR, 1900 ms; flip angle, 150 degrees; thickness, 1 mm; FOV, 250 mm × 250 mm). Between November 2014 and November 2023, all newborn brain MRI scans in our department were obtained on this machine according to the above scanning parameters.

### 2.4. MR Image Analysis

#### 2.4.1. Measurement of the Craniocortical Width (CCW)

Bilateral CCW was measured on the coronal plane at the level of the foramen of Monro, shown in the first image of Figure 2; A CCW ≤ 4 mm was defined as normal, >4 but ≤5 mm was defined as mild ESS, and >5 mm as severe ESS [4,11]. To validate the method, brain metrics were measured in all enrolled cases by two experienced investigators blinded to the outcome data. CCW was measured twice independently by both investigators. Intra-observer agreement was excellent with an Intra-class Correlation Coefficient (ICC) > 0.93 for all measurements.

#### 2.4.2. Brain Tissue Segmentation

At present, there is no software platform that can directly segment CSF in the extra-cerebral space, and the segmentation needs to be obtained indirectly. Our strategy is to integrate the respective advantages of two skull-stripping tools, Brain Surface Extractor (BSE) (Version 2.1) and SynthStrip (Version 1.0). BSE can obtain an accurate mask of total CSF in the cranial cavity, while Sythstrip can obtain an accurate mask of brain tissue containing CSF in the ventricular system.

To obtain the exact volume of the outer space, the inner edge of the cranial cavity needs to be acquired, so we use a semi-automatic method to segment the whole brain. The automatic software for neonate brain segmentation will underestimate the mask of the outer space [12]. We took the T1-mpr images as primary data; then, we put Neuroimaging Informatics Technology Initiative (NIFTI) files translated from primary T1-mpr images into the Brain Surface Extractor (BSE) to carry out the skull-stripping [13]. In case of over-stripping of the CSF, we manually adjusted the brain mask which was obtained automatically through BSE.

After obtaining the skull-stripped brain volume with the accurate edge of CSF, we put those images to FSL, which can automatically segment the brain via FMRIB’s Automated Segmentation Tool (FAST) (Version 6.0) into three tissues: gray matter (GM), white matter (WM), and CSF [14]. The volume of the intracranial cavity (ICC) is the sum of volumes of GM, WM, and CSF, and the volume of brain parenchyma was the sum of GM and WM. To obtain the volume of CSF in the ventricular system, we used the data obtained from Infant FreeSurfer’ ROI analysis [15] because the FSL cannot calculate this index. The volume of extra-CSF was obtained by subtracting the volume of CSF in the ventricular system from the volume of total CSF, as shown in Figure 2. All segmentation results need to undergo visual inspection. Cases with poor segmentation results can be re-segmented by adjusting FAST parameters. The process of skull-stripping and brain tissue segmentation for all cases was supervised by a radiologist with 20 years of experience.

#### 2.4.3. Brain Surface Segmentation

The infant FreeSurfer software (https://surfer.nmr.mgh.harvard.edu/fswiki/infantFS, accessed on 1 January 2023) was used to automatically process and reconstruct the cortical surface from T1-mpr images. Infant FreeSurfer is designed specifically for infants aged 0–2 years [15]. To carry out skull-stripping, a tool named SynthStrip was used, which leverages a deep learning strategy to synthesize arbitrary training images from segmentation maps [16]. The result of the segmentations was visually inspected and manually edited via Itksnap (Version 3.8.0) in case of minor voxel misclassifications [17]. Seven images were discarded due to parcellation errors. Four key indices of cortex maturation, including surface area, cortical volume, cortical thickness, and mean curvature (MC), were fetched from the infant FreeSurfer’s output. The route diagram of MRI post-processing is shown in Figure 2.

### 2.5. Statistical Analysis

Data were analyzed using SPSS version 22.0 software. Analysis of covariance (ANCOVA) was used for continuous parametric variables (birth weight, gestational age, postmenstrual age at MRI scan, and volumes of brain tissues). Chi-squared tests were used for categorical variables (IVH, PWML, sBPD, IUGR, and postnatal steroid usage). A *p* value < 0.05 indicates statistical significance. Surface-based group analysis was used to compare the four indices of cortex development between the three groups. The general linear model approach was carried out using Freesurfer software, to perform a vertex-wise analysis across the whole brain to examine differences in cortical thickness, surface area, cortical volume, and mean curvature. The linear model was fitted using a Different Onset Different Slope (DODS) method and corrected GA at scan and GA were brought into the model as demeaned covariates. A false discovery rate (FDR) was then applied to correct for multiple comparisons. The resulting difference maps show statistically significant differences if they survive an FDR correction of 0.05. Clusters of significant vertex-base changes were labeled with reference to the Desikan–Killiany atlas.

## 3. Results

### 3.1. Participant Characteristics

A total of 200 infants were finally enrolled in this study. Of these, 90 infants (45%) were diagnosed with ESS, while the severe-ESS group accounted for 24.4% of the total cases with ESS, and the information about stratification of GA in each group was showed in Figure 3. Because many infants were excluded from our study, the above percentage cannot represent the real incidence of ESS.

### 3.2. Comparison of Basic Clinical Characteristics Between the Three Groups

The severe-ESS group had a lower GA and birth weight than the mild-ESS and no-ESS groups, but the difference was not statistically significant. The average corrected GA at MRI scan among the three groups was around 39 weeks. The head circumference (HC) at MRI scan in the groups with ESS was significantly larger than in the group without ESS. Perinatal factors related to brain volume change, such as antenatal and postnatal steroid use, IUGR, and sBPD, did not differ. There was no significant difference between the three groups regarding the brain injuries, such as IVH and WMI, as shown in Table 1.

### 3.3. Comparison of Brain Volume Between Three Groups

The segmented volumetric results of the three groups are shown in Table 2. The volume of ICC and total CSF were significantly higher in the two groups with ESS; the volume of CSF in the severe-ESS group was almost two times as high as that of the no-ESS group. We did not find any differences in the volume of CSF filled in the ventricular system between the two groups with ESS, but the severe-ESS group had bigger ventricles than the no-ESS group. The absolute volume and percentage of gray matter and white matter in the brain did not differ between the mild-ESS group and the no-ESS group, while the severe-ESS group had smaller brain parenchyma than the no-ESS group, as shown in Table 2.

### 3.4. Comparison of Indices of Cortex Maturation Among Three Groups

There was no significant difference in surface area, cortical volume, and cortical thickness between the three groups. As shown in Figure 2, the significant clusters can be seen when comparing the mean curvature between the groups with ESS and the group without ESS. The group with severe ESS had lower mean curvature in the regions around the frontal and parietal lobes of both hemispheres than the group without ESS. The mild-ESS group also had the same trend compared with the no-ESS group, but the regions involved changed a little. Details about the regions with significant differences are shown in Table 3 and Figure 4 and Figure 5.

## 4. Discussion

To the best of our knowledge, this is the first work to analyze the associations of mild and severe ESS with brain volume and cortical folding changes in very preterm infants using 3D reconstructed MRI. The current study showed that severe ESS may be a new risk factor for limited brain growth in preterm infants and that the enlargement of extra brain space could alter the morphology of the cortex during the early stages of life.

### 4.1. ESS and Brain Volume

This study demonstrated that the ESS was associated with the brain growth in preterm infants, but only in infants with severe ESS. This finding can partially answer the question of the relationship between ESS and adverse neurodevelopmental outcomes, because smaller brain parenchyma usually means poorer neurodevelopmental performance. Many perinatal events have been proven to be risk factors for decreasing brain volume in preterm infants, such as BPD, IUGR, postnatal steroid use, and IVH, as well as PWML [8,18]. In this study, the difference in these factors was shown to be not significant, which means that severe ESS itself may be recognized as a new risk factor for brain growth in the early life stages of preterm infants. This finding indicates that neonatologists should pay more attention to these significant differences related to severe ESS in clinical work. On the other hand, we found that mild ESS did not have the same associations, which means that the majority of ESS was spared regarding brain parenchyma volume, and this finding assures us that the existence of ESS at the time of discharge was a very common and safe phenomenon in very preterm infants.

The different associations with brain volume between severe ESS and mild ESS tell us that the “trade-off” theory may not be as tenable as some studies expected. A study from Boardman JP found that the preterm infants had more volume of CSF than term infants, and they explained this phenomenon with abnormal white matter development [19]. Another similar study by Kelser SR also illustrated that the expansion of cerebrospinal fluid space was secondary to subcortical tissue loss. The key point of this theory is that the change in CSF volume should be connected firmly with the alteration of brain parenchyma [20]. Our research findings indicate that not all increases in cerebrospinal fluid volume are accompanied by a decrease in brain parenchyma; therefore, more theoretical support is needed. When we look at the normal trajectory of change in the subarachnoid space since the fetus stage, the volume of extra-CSF continuously increases with GA until it reaches the peak volume at around 36 weeks (160 mL); then, it decreases gradually. At 39 weeks of CGA, the volume of extra-CSF drops to around 40 mL, which is the quarter of the volume at 36 weeks [21,22,23]. In our study, the average volume of extra-CSF was only around 60 and 80 mL in the mild- and severe-ESS groups, respectively, while the difference in brain volume between the two groups was very small. This means that the driving factor of ESS may not only be caused by the loss of other structures in the cranial cavity but by CSF itself; whether there is overproduction or malabsorption of CSF, any impairment on the whole process of CSF circulation could cause changes. Based on normal brain-CSF biomechanics [24,25], we speculate that there is a balance between the distribution of extra-CSF and brain parenchyma; when the subarachnoid space expands too much in a short period of time, the brain growth might slow down accordingly. Another possible explanation for the results of this study is that the underlying factors in the severe-ESS group affected brain growth and CSF circulation at the same time, and further studies are needed to support this explanation.

Compared with other perinatal factors associated with changes in brain volume [8], the degree of brain volume loss in preterm infants with severe ESS was actually mild. The gap of brain volume between severe ESS and no ESS was only 3.2%. The increase in ICC and HC was more obvious than the loss of brain parenchyma in the severe-ESS group; this means that the cavity tissue is softer than the brain parenchyma, and HC may not accurately represent real parenchyma volume in preterm infants with ESS. HC has been considered as a classic index to evaluate the brain growth in preterm infants for a very long time, because many studies have proven that HC has a very strong relationship with brain volume, and a larger HC means better brain development [11]. Our study theorizes a different idea that, under the existence of ESS, a larger HC may not necessarily represent better brain growth in preterm infants.

### 4.2. ESS and Cortex Development

The evidence from the change in cortical morphology also supports that the abnormal accumulation of CSF in the subarachnoid space may have a detrimental effect on brain maturation in preterm infants. Our study showed that both mild and severe ESS can decrease the mean curvature of the cortex. Mean curvature is one of the key indices that can represent cortex maturation in early life. Like the local gyrification index, the value of MC contains information about cortex folding, which can be influenced by many perinatal factors [26,27]. Compared with term infants, preterm infants usually have a higher MC at TEA, which means that the difference between the environment inside and outside the uterus can affect the development of the cortex [28,29]. The authors explained that preterm infants tend to have shallower sulci than term infants, and these sulci retain high-curvature troughs but lack low-curvature sulcal wall area, leading to higher overall curvature. However, in our study, MC was observed to have a downward trend, which is opposite to the effect of preterm birth, but similar to brain injury. Joshua S. Shimony compared the difference in effect on folding measures between preterm infants with and without brain injury. The results showed that the absolute MC of the group with brain injury at TEA was lower than that of the group without brain injury [30], showing that ESS could affect cortex development similarly to brain injury. However, we believe that the mechanism of action is totally different.

We thought that the pressure from ESS could compress the surface of cortex, making the gyrus flatter and the sulcus shallower, thus resulting in a decrease in MC [30]. In support of this argument, we found that the affected regions were similar to those places adjacent to more obvious ESS, such as bilateral temporal and frontal lobes. However, we did not find any difference in the cortical thickness, which was expected before the study, showing that the influence of ESS on cortical development may be small. Cortical folding mechanisms are not yet fully understood [27]; regarding biomechanics, the hypothesis that the skull and meninges constrain the outer expansion of the cortex has been a popular opinion, but it has been disproved by experiments based on animal models [31]. Our findings based on preterm infants still support this idea as well as the previous hypothesis: the pressure from outside the brain can cause some changes in cortex morphology.

Regarding prognostic capacity, MC at TEA is weaker than the surface area and gyrification index [30,32,33], and the clinical meaning of reduced MC in preterm infants remains unclear [34,35]. Julia E Kline found that curvature of the inner cortex in the insula and temporal lobes at TEA was negatively correlated with cognitive and language scores at 2 years [29,36]. Lower MC at TEA seems to be a protective factor for long-term neurodevelopment. However, it should be noted that “reduced MC” does not necessarily represent “lower MC”, and the real level of MC in our study could still be located in a high-level range, because our research subjects are all preterm infants who are expected to have relatively higher MC value at TEA. Moreover, a study focusing on the gyrification of premature-born adults showed that, compared with term-born adults, the absolute MC was highly increased at 27 years of age and was positively associated with brain injury at birth [37]. However, as mentioned above, brain injury can cause a decrease in MC at TEA [30], meaning reduced MC at TEA may reverse to increased MC in adulthood. In addition, a study found that reduced MC at TEA was associated with a common genetic variation related to autism spectrum disorder and lipid metabolism [38]. Overall, the reduced MC alone shows a limited predictive capacity and slightly tends towards negative impacts. The question of whether ESS was related to adverse long-term neurodevelopmental outcomes cannot be fully answered by the changes in MC seen in this study, and further longitudinal studies on reduced mean curvature in preterm infants with ESS are needed.

## 5. Limitations

When interpreting the results of our study, several limitations need to be considered. First, this is a retrospective single-center study, which means the number of enrolled babies was relatively low, and our cohort excluded babies with severe brain injury and babies without 3D-MRI; these factors limit our analysis of the incidence of ESS and associated risk factors. Second, we did not consider the exact onset time of ESS, which is important to evaluate the overall effect of ESS on brain growth. Theoretically, a longer existence of ESS leads to a more significant change, but a prospective study is needed to conclude whether this assumption is tenable. Third, the scan time of brain MRI was variable, and this may introduce obvious bias in the accuracy of evaluation of brain volume and cortex development because the period encompassing the last trimester of pregnancy and early postnatal life is characterized by dynamic progressive events that shape the developing cortex. In our study, we considered a CGA of 37–40 weeks as the range of enrolment, and the results showed that the average scan time among the three groups was similar, even though the range of scan time still covered 3 weeks; stratification analysis by scan time should be performed if there are sufficient samples in each layer of GA. Fourthly, our study lacks neurodevelopmental follow-up data, which is crucial for characterizing ESS. Evidence from the positive results of this study can only establish an association between ESS and abnormal quantitative indices of brain maturation. Finally, regarding controlling confounders, there is no doubt that the GA can have a huge impact on brain growth and cortex maturation, but owing to balanced distribution of different GA in the three groups, we did not perform a stratification analysis according to GA, and this could cause some bias in the results. Apart from the confounding factors listed in Table 1, it is likely that some unmeasured factors can affect the volume of brain tissues and cortex morphology, and bias may be caused in this study by these uncontrolled factors.

## 6. Conclusions

Using advanced MRI and surface-based analysis, this study showed that ESS can be taken as a potential biomarker of impaired brain development early in life in preterm infants, which includes decreased brain parenchyma volume and decreased cortical mean curvature. This study highlights that severe ESS may be a risk factor for brain volume loss in very preterm infants. ESS may also change the degree of cortex curvature, but the effect of this change on neurodevelopmental outcomes is unclear. The findings of this study cannot fully answer how ESS is related to adverse neurodevelopmental outcomes in preterm infants, but they shed light on the fact that ESS in preterm infants may not be benign and further research is necessary, especially longitudinal RCT studies with larger samples.

## Figures and Tables

**Figure 1 diagnostics-15-02206-f001:**
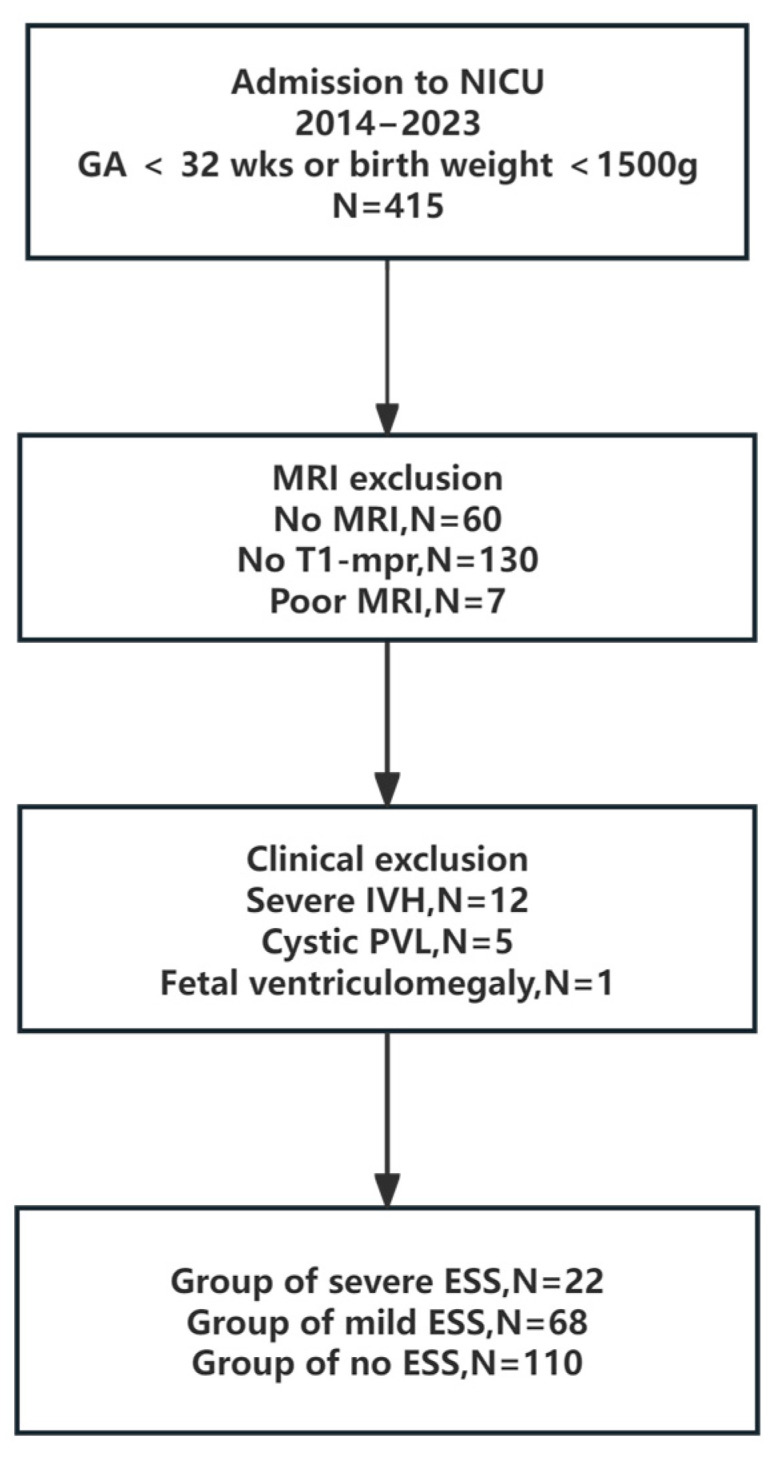
Flowchart of the study design. NICU—neonatal intensive care unit; IVH—intraventricular hemorrhage; cystic PVL—cystic periventricular leukomalacia; MRI—magnetic resonance imaging; Wks—weeks; ESS—enlarged subarachnoid space.

**Figure 2 diagnostics-15-02206-f002:**
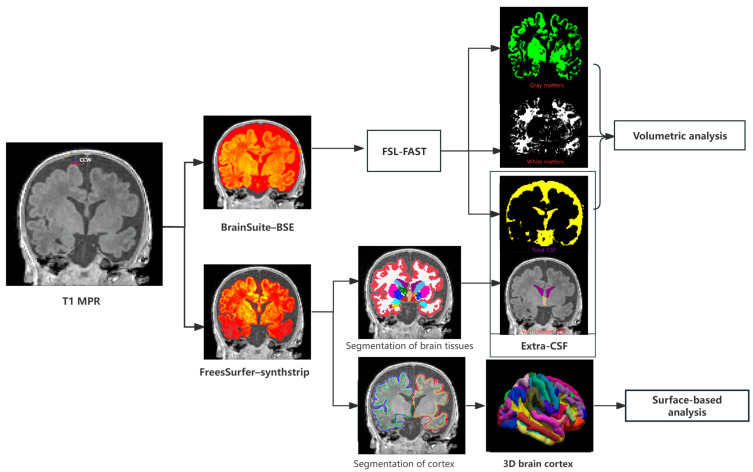
Simplified overview of the neonatal pipeline for magnetic resonance imaging processing. CCW was measured on the coronal plane at the level of the foramen of Monro. Two types of skull-stripping were performed with BSE and Synthstrip in volumetric analysis and surface-based analysis, respectively. The volume of extra-CSF was obtained by subtracting the volume of CSF in the ventricular system from the volume of total CSF obtained from the results of segmentation in FSL and infantFreeSurfer, respectively. CCW—craniocortical width.

**Figure 3 diagnostics-15-02206-f003:**
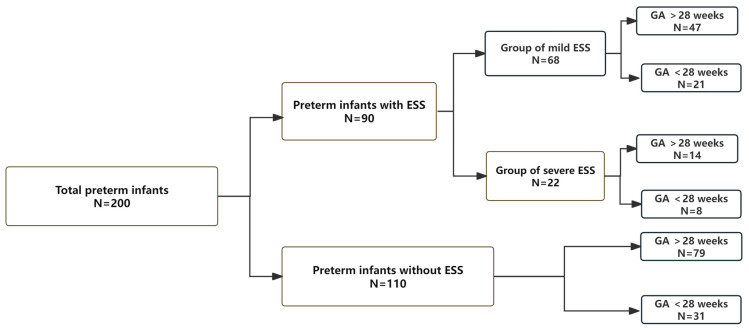
Flowchart of group stratification. ESS—enlarged subarachnoid space; GA—gestational age.

**Figure 4 diagnostics-15-02206-f004:**
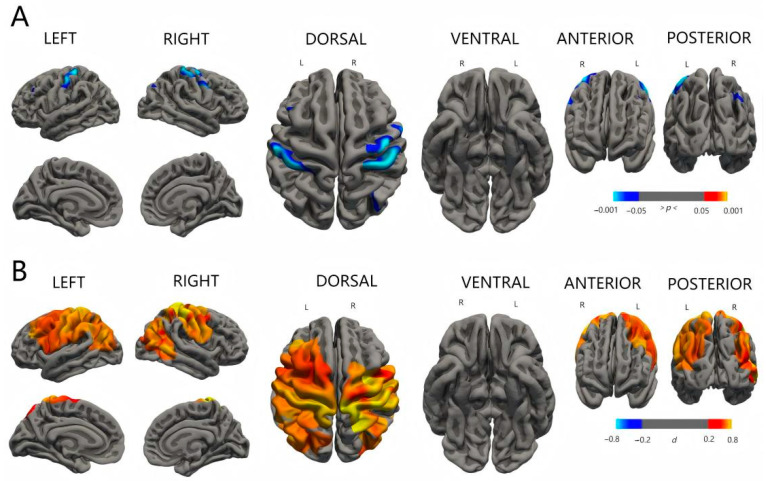
Differences in cortical mean curvature between mild-ESS group and no-ESS group. Row A shows *p*-maps and row B shows effect size. The *p*-maps were produced from GLM models fitted at each vertex across the cortical surface, with cortical curvature as the dependent variable and group as the independent variable, co-varying for gestational age at birth. The threshold of the *p*-maps was set to yield an expected 5% FDR across both hemispheres. In the effect size maps, red–yellow color represents areas of decreased mean curvature. Abbreviations: *d*: Cohen’s *d*; FDR: false discovery rate.

**Figure 5 diagnostics-15-02206-f005:**
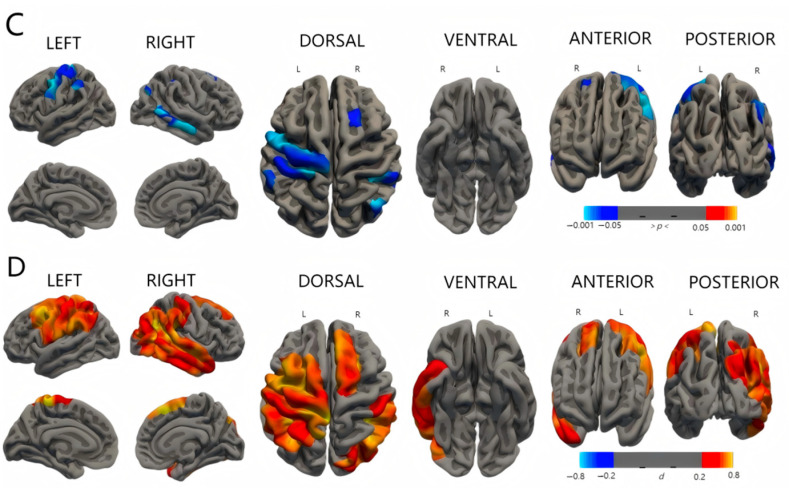
Group differences in cortical mean curvature between severe-ESS group and no-ESS group. Row C shows *p*-maps and row D shows effect size. The *p*-maps were produced from GLM models fitted at each vertex across the cortical surface, with cortical curvature as the dependent variable and group as the independent variable, co-varying for gestational age at birth. The threshold of the *p*-maps was set to yield an expected 5% FDR across both hemispheres. In the effect size maps, red–yellow color represents areas of decreased mean curvature. Abbreviations: *d*: Cohen’s *d*; FDR: false discovery rate.

**Table 1 diagnostics-15-02206-t001:** Perinatal characteristics of study population.

	No ESS (n = 110)	Mild ESS (n = 68)	Severe ESS (n = 22)	*p* Value
Birth weight (g)	1201 ± 276	1125 ± 279	1078 ± 245	0.798
Gestational age (weeks)	29.3 ± 2.5	28.5 ± 2.6	28.1 ± 2.4	0.522
Head circumference at birth (cm)	26.1 ± 2.2	26.3 ± 2.4	25.6 ± 1.8	0.854
Head circumference at MRI scan (cm)	32.5 ± 0.9 ^1,2^	33.6 ± 1.0 ^1,3^	33.5 ± 1.2 ^2,3^	***p* = 0.001 ^1^** ***p* = 0.001 ^2^** *p* = 0.921 **^3^**
Gestational age at MRI scan (weeks)	38.0 ± 2.3	38.2 ± 2.1	38.5 ± 2.5	0.588
Male sex	64 (58.2)	38 (55.9)	13 (59.0)	0.885
Grade 1–2 IVH	14 (12.7)	7 (10.3)	4 (18.2)	0.459
PWML	8 (7.3)	4 (5.9)	3 (13.6)	0.631
sBPD	80 (72.7)	38 (55.9)	14 (63.6)	0.954
IUGR	24 (21.8)	15 (22.0)	6 (27.3)	0.559
Postnatal steroid usage	6 (5.5)	4 (5.9)	2 (9.0)	0.452

MRI—magnetic resonance imaging; IUGR—intrauterine growth restriction; IVH—intraventricular hemorrhage; PWMLs—punctuate white matter lesions; sBPD—severe bronchopulmonary dysplasia. Bold values indicate significance at *p* <0.05.The superscript number 1 represent the comparision between the No ESS and the Mild ESS; The superscript number 2 represent the comparision between the No ESS and the Severe ESS;The superscript number 3 represent the comparision between the Mild ESS and the Severe ESS.

**Table 2 diagnostics-15-02206-t002:** Comparison of volumetric indices between three groups.

				Pairwise Comparison *p* Value		
	No ESS (n = 110)	Mild ESS (n = 68)	Severe ESS (n = 22)	Mild vs. No ESS	Severe vs. No ESS	Severe vs. Mild ESS
Volume of ICC (mL)	356.78 ± 26.03	374.25 ± 26.45	384.66 ± 30.33	** *p* ** **= 0.000**	** *p* ** **= 0.001**	*p* = 0.291
Volume of brain parenchyma (mL)	312.27 ± 20.75	310.37 ± 24.41	302.35 ± 26.43	*p* = 0.426	** *p* ** **= 0.003**	** *p* ** **= 0.001**
% ICC	87.52 ± 3.1	82.93 ± 2.7	78.60 ± 4.6	** *p* ** **= 0.013**	** *p* ** **= 0.000**	** *p* ** **= 0.010**
Total volume of CSF (mL)	44.51 ± 8.9	63.88 ± 7.4	82.31 ± 8.2	** *p* ** **= 0.000**	** *p* ** **= 0.000**	** *p* ** **= 0.000**
% ICC	12.47 ± 2.2	17.06 ± 2.9	21.39 ± 3.7	** *p* ** **= 0.013**	** *p* ** **= 0.000**	** *p* ** **= 0.010**
Volume of CSF in ventricular system (mL)	6.41 ± 1.8	8.52 ± 2.1	10.33 ± 2.5	*p* = 0.298	*p* = 0.336	*p* = 0.181
% ICC	1.79 ± 0.7	2.27 ± 0.5	2.68 ± 0.4	*p* = 0.743	*p* = 0.063	*p* = 0.077
Volume of extra CSF (mL)	40.54 ± 4.3	55.36 ± 3.8	74.20 ± 5.1	** *p* ** **= 0.000**	** *p* ** **= 0.000**	** *p* ** **= 0.000**
% ICC	11.36 ± 1.9	14.79 ± 1.4	19.29 ± 2.0	** *p* ** **= 0.009**	** *p* ** **= 0.000**	** *p* ** **= 0.027**

Bold values indicate significance at *p* < 0.05.

**Table 3 diagnostics-15-02206-t003:** Whole-brain-surface-based analysis results with significant clusters set at *p* < 0.05.

	Severe ESS/No ESS					
	Annotation	Cluster Size (mm^2^)	MNI			Sig.
			x	y	z	
**Mean Curvature**	lh_precentral	985.35	−43.79	−11.40	61.76	0.001
	lh_postcentral	401.60	−47.91	−31.23	53.79	0.012
	lh_supramarginal	105.11	−57.83	−44.76	44.87	0.025
	rh_middletemporal	770.56	64.88	−53.12	4.10	0.003
	rh_inferiorparietal	201.43	51.32	−55.63	47.12	0.010
	rh_supramarginal	267.14	60.31	−33.63	45.10	0.005
	rh_caudalmiddlefrontal	355.88	39.54	27.08	44.73	0.032
	**Mild ESS/No ESS**					
	**Annotation**	**Cluster Size** **(mm^2^)**	**MNI**			**Sig.**
			**x**	**y**	**z**	
	lh_postcentral	675.44	−35.00	−33.03	69.25	0.001
	lh_caudalmiddlefrontal	140.98	−42.69	13.34	52.10	0.002
	rh_precentral	765.33	43.01	−7.84	60.92	0.036
	rh_postcentral	512.43	38.19	−31.29	68.28	0.008
	rh_inferiorparietal	266.34	54.67	−48.01	39.42	0.040

Note: MNI (Montreal Neurological Institute) indicate MNI305 coordinates.

## Data Availability

Data are available from the corresponding author upon reasonable request.

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
