# Peer review of "The Impacts of Enlarged Subarachnoid Space on Brain Growth and Cortex Maturation in Very Preterm Infants"

_diagnostics, 2025, doi:10.3390/diagnostics15172206_

Round 1
Reviewer 1 Report
Comments and Suggestions for Authors
Dear Authors,
I have read an article entitled "The impacts of enlarged subarachnoid space on the brain growth and cortex maturation in very preterm infants". Several points need to be addressed:
1) The authors need to be consistent with abbreviations. They should be lengthened before being abbreviated. With that being said, what do MDI and PDI stand for?
2) "This study aims to explore the associations of ESS with brain growth and cortex maturation.." The aim seems to be off. By reading this aim, I expect the research design to be of a longitudinal design if brain growth and cortex maturation were to be assessed; instead, it is just a cross-sectional study.
3) How was sampling conducted?
4) More importantly, what were the criteria for no, mild and severe ESS?
5) There are no relevant clinical data to merit the conclusion of "..associated with adverse brain development early in life in preterm infants" as all of the measurements are just radiological indices.
Reviewer 2 Report
Comments and Suggestions for Authors
Review of the Manuscript: “The impacts of enlarged subarachnoid space on the brain Growth and cortex maturation in very preterm infants”
This manuscript presents a retrospective cohort study assessing the connection between enlarged subarachnoid space (ESS) and brain growth and cortical development in very preterm infants using advanced MRI-based volumetric and surface analysis. The Authors present fascinating evidence that severe ESS is associated with decreased brain parenchyma volume and changes in cortical curvature, particularly in the parietal and temporal lobes, which may indicate early-life alterations in brain maturation.
Overall, the study reports an important and underexplored clinical question with potential relevance for neonatal care. The manuscript is rich in methodological detail and supported by quantitative MRI data. However, several clarifications and improvements are recommended to enhance clarity, rigor, and clinical contextualization.
Title and Abstract
-The title is clear, but the abstract should more explicitly highlight the retrospective design and potential limitations associated with selection bias
-the Authors should rephrase the conclusion in the abstract to emphasize that ESS “may be associated” with adverse outcomes, rather than proposing causality.
Introduction
-Condense background on “trade-off theory” to improve focus
- Explain what abbreviations MDI and PDI refers to (line 39)
-Pay attention to punctuation (e.g. add a space before and after brackets, lines 47 and 48) and line 53
-Add 1–2 lines on evidence gaps in ESS-related neurodevelopmental outcomes beyond age two
-Clarify the novelty of this study (e.g., segmentation strategy, curvature analysis).
Methods
-Justify the retrospective design and describe how imaging quality was ensured over time
-The Authors should consider a classification of outcomes based on gestational age
-Regarding brain tissue segmentation, clarify why both BSE and SynthStrip were used; describe segmentation error management.
Results
-Include absolute values with percentages
-As indicated in the methods section, add a table regarding the classification of results based on gestational age
-Present curvature metrics with effect sizes (if possible) to support interpretation.
Discussion
-Avoid causal language (e.g., modify “ESS impairs” to “ESS is associated with”)
-Add relationships with present literature es.” The different associations on brain volume between severe ESS and mild ESS told us that the “trade-off” theory may not be as tenable as some study expected.(lines 240-241); contextualize with similar imaging studies in preterms
-Pay attention to punctuation and use of capital letter (line 253)
-Discuss more in depth the prognostic implications of reduced mean curvature, referencing longitudinal outcome studies.
Limitations
-Expand on:
- Lack of neurodevelopmental follow-up data
- Confounders like GA variability and unmeasured factors
- MRI positioning effects on CCW/curvature.
Language and Style
-Ensure clear, consistent phrasing.
-Avoid ambiguous terms like “impaired brain”
-Reduce jargon where possible
-Revise punctuation in all the manuscript.
Figures and Tables
-Add a visual group stratification flowchart early in the sections Results.
Conclusion
-Frame ESS as a potential biomarker, not a causal agent.
Round 2
Reviewer 1 Report
Comments and Suggestions for Authors
The authors have addressed almost all of my questions.
There is just one issue, where if the drawing made in the response to authors regarding how to measure ESS is not copyrighted, please include them in the manuscript as it will improve reproducibility.
